# OCTAVVS: A Graphical Toolbox for High-Throughput Preprocessing and Analysis of Vibrational Spectroscopy Imaging Data

**DOI:** 10.3390/mps3020034

**Published:** 2020-05-01

**Authors:** Carl Troein, Syahril Siregar, Michiel Op De Beeck, Carsten Peterson, Anders Tunlid, Per Persson

**Affiliations:** 1Department of Astronomy and Theoretical Physics, Lund University, 223 62 Lund, Sweden; syahril.siregar@thep.lu.se (S.S.); carsten@thep.lu.se (C.P.); 2Department of Biology, Lund University, 223 62 Lund, Sweden; michiel.op_de_beeck@biol.lu.se (M.O.D.B.); anders.tunlid@biol.lu.se (A.T.); per.persson@biol.lu.se (P.P.); 3Centre for Environmental and Climate Research (CEC), Lund University, 223 62 Lund, Sweden

**Keywords:** infrared spectroscopy, hyperspectral, atmospheric correction, Mie scattering correction, MCR-ALS

## Abstract

Modern vibrational spectroscopy techniques enable the rapid collection of thousands of spectra in a single hyperspectral image, allowing researchers to study spatially heterogeneous samples at micrometer resolution. A number of algorithms have been developed to correct for effects such as atmospheric absorption, light scattering by cellular structures and varying baseline levels. After preprocessing, spectra are commonly decomposed and clustered to reveal informative patterns and subtle spectral changes. Several of these steps are slow, labor-intensive and require programming skills to make use of published algorithms and code. We here present a free and platform-independent graphical toolbox that allows rapid preprocessing of large sets of spectroscopic images, including atmospheric correction and a new algorithm for resonant Mie scattering with improved speed. The software also includes modules for decomposition into constituent spectra using the popular Multivariate Curve Resolution–Alternating Least Squares (MCR-ALS) algorithm, augmented by region-of-interest selection, as well as clustering and cluster annotation.

## 1. Introduction

Hyperspectral images, i.e., images where each pixel contains continuous spectral information, are important in a wide field of sciences and applications, and across a wide range of spatial scales. Several techniques are available for collecting hyperspectral images, and together, these cover a large part of the electromagnetic spectrum. Infrared (IR) spectroscopy has become popular partly because it provides label-free measurement of the distribution and dynamics of inorganic and organic compounds in many different types of samples in biological, chemical, environmental, geological and physical sciences. This technique exploits vibration modes of chemical bonds that give different classes of molecules distinct spectra in the IR region, making vibrational spectroscopy a powerful tool for investigating the molecular composition of complex biological samples, applicable at many different length scales.

Detector development has enabled the collection of spatially resolved IR spectra, which is particularly useful for the study and classification of biomedical tissues [1]. Since their commercial introduction in the late 1990s, IR detectors have evolved to rapidly record thousands of spectra with micrometer spatial resolution, comparable to the IR wavelengths. Experiments generate large amounts of data as the resolution and image count increase; the storage of a few hundred images may require a hundred gigabytes or more.

Various algorithms have been developed to handle experimental artifacts commonly encountered in IR microspectroscopy, such as atmospheric contributions to the spectra [2,3], Mie scattering of photons by structures on the μm scale [4,5,6,7], and shifts in spectral baselines caused by unspecified effects. Following such preprocessing steps, spectra may be analyzed, visualized and reduced by feature extraction methods; the principal component analysis (PCA) is often used to extract the main features of the data. Many scientific questions require the application of unsupervised or supervised classification, e.g., to segment individual images or to distinguish healthy from diseased cells [1].

In principle, a researcher can employ a suitable combination of published software packages and algorithms to correct and analyze large sets of hyperspectral data. In practice, a considerable amount of work may be needed to find and adapt existing tools and methods, often in different programming languages, to work together as a data pipeline. To make full use of experimental data, researchers need software that allows non-experts to quickly and easily process spectra and extract scientific results.

IR spectrometers are generally accompanied by software for preprocessing and analysis of spectra; examples include SpectrumIMAGE from PerkinElmer, OPUS from Bruker and Resolutions Pro from Agilent. While useful for many tasks, these applications are proprietary, platform-specific software whose inner workings are not revealed. Some, such as OPUS, offer extensive scripting options, but there is no standard language or interface across these different hardware and software platforms. Instead, data may be exported for further processing in external tools.

There are also several free software packages for working with spatially resolved vibrational spectra. HYPER-Tools is a graphical application for MATLAB, with a range of commonly used preprocessing options and analysis algorithms [8]. The user interface allows different methods to be explored, but programming skills are needed to load and format the data matrices, and to create processing pipelines. In contrast, complex workflows with analysis and visualization can be drawn in the Orange visual programming toolkit [9] with the Quasar spectroscopy extension [10], which supports many file formats and algorithms specific to spectroscopy. As free and open-source software, Quasar can readily be extended with new methods, though the complexity level is relatively high.

Working with large data sets in a cross-disciplinary environment, we have observed a need for user-friendly graphical tools that allow researchers to process their data easily and efficiently while operating on whole data sets, not only individual images. Modularity and flexibility are often important, as the type of data and scientific questions determine what individual steps and procedures are appropriate, but there is also value in defining sensible pipelines for users.

In response to these challenges, we created a set of graphical tools for performing preprocessing and clustering of hyperspectral images, primarily from IR microspectroscopy. We focused on three key areas where further method development and adaptation were needed; these are introduced below. The software itself is described in Results.

### 1.1. Atmospheric Correction

Vibrational excitation of water vapor gives rise to numerous fine absorption lines in two distinct regions, around approximately 1300–2000 cm${}^{-1}$ and 3500–4000 cm${}^{-1}$. The former region is the more troublesome, as it coincides with the main peaks of important organic molecules, including the Amide I peak of proteins. Similarly, carbon dioxide causes a broad double peak around 2300–2440 cm${}^{-1}$, as well as peaks at 600–700 cm${}^{-1}$ [2]. Background subtraction is incapable of fully removing these unwanted spectral contributions, due to temporal and spatial variation in the levels of the unwanted atmospheric gases. Instead, methods have been devised for atmospheric correction of individual spectra, based on subtracting a suitably scaled atmospheric reference spectrum for each affected region.

### 1.2. Mie Scattering Correction

Biological samples contain structures such as cells and organelles of sizes comparable to the IR wavelengths, leading to scattering effects that distort spectra by altering intensities in a wavenumber-dependent way. This is particularly a concern for transflection measurements, where resonant Mie scattering adds non-constant baselines [1]. Kohler et al. first used extended multiplicative signal correction (EMSC) to model and remove scattering effects generated by particles of unknown sizes and optical properties [4]. The resonant Mie scattering correction algorithm (RMieS-EMSC) by Bassan et al. also takes into account the link between refraction and absorption through the Kramers–Kronig relations, leading to an iterative procedure where a reference spectrum is successively refined [11]. Application of RMieS-EMSC to imaging data has demonstrated its power as a method for extracting spectral information amenable to classification [12]. However, the method takes several seconds to correct a single spectrum, requiring supercomputers to process large data sets. Algorithmic improvements were introduced by Konevskikh et al. to reduce the computational complexity [6,13]. These improvements became accessible to programmers with the recent publication of the ME-EMSC algorithm by Solheim et al. [7]; the ME-EMSC implementations in MATLAB and, more recently, Python provide a good reference point for further development of the method.

### 1.3. Factorization and Segmentation

PCA is commonly used to find a low-dimensional projection of high-dimensional data. Due to the inherent non-negativity of both spectra and chemical concentrations, non-negative matrix factorization is a more natural choice than PCA for decomposing hyperspectral images; the resulting bilinear model describes the spectral data in terms of concentrations of a small set of putative pure spectra. The aim of this broad family of methods is generally both to reduce the dimensionality of data and to reveal physically meaningful patterns that may not otherwise be readily visible [14]. The Multivariate Curve Resolution (MCR) framework is commonly used in spectroscopic data analysis, but also in other fields where multiple components have to be identified from unknown mixtures, in particular chemical process analysis [15].

In the MCR-ALS method, MCR is coupled with alternating least squares (ALS) optimization: the spectra and concentrations are alternatingly updated, with constraints such as non-negativity enforced at each update. Advantages of MCR-ALS include its ability to obtain information from multiple data matrices [16,17] and its ability to deal with many types of constraints to improve the factorization results [18]. The method has been applied to data from a wide range of spectroscopic techniques such as IR [19,20], Raman [21], UV–visible [22], fluorescence [23] and NMR [24] spectroscopy. It should be noted that the decomposition can be degenerate due to rotations, but the output of MCR-ALS depends also on specifics of the optimization method [25].

A widely used tool for MCR-ALS analysis is the MCR-ALS GUI [26], an open-source MATLAB program that offers many options for algorithms and constraints. Derived from this is a more specialized tool for analysis and subsequent clustering of spectroscopic images, also referred to as the MCR-ALS GUI [19]. While these are powerful tools for analyzing single images, their graphical interfaces are not designed for processing large sets of spectroscopic images. As a workaround, multiple images may be stitched together into one large image, limited in size by memory requirements.

## 2. Results

We developed an operating system-independent graphical toolbox for processing large sets of hyperspectral images: Open Chemometrics Toolbox for Analysis and Visualization of Vibrational Spectroscopy data, OCTAVVS (pypi.org/project/octavvs). OCTAVVS is written in Python 3, using PyQt5 for the graphical user interfaces and Matplotlib for drawing graphs. A diagram showing the data flow through the software is presented in Figure 1. Representative screenshots are shown in Figure 2 and Figure 3.

As the software is geared towards rapid processing of large data sets, both the preprocessing module (Figure 1, left) and the MCR-ALS and clustering analysis modules (Figure 1, right) can work with whole sets of images. The user may load one or more images from files in MATLAB, OPUS or CSV formats, select what processing steps to apply, adjust parameters in response to their effect on an image or a few selected spectra, and finally run the data processing on the full set of images sequentially.

In all three software modules, several small graphs show the result of the processing steps that have been applied, and these graphs may be enlarged and saved in different image formats. The current image is visualized as a heat map, based on the total or maximal intensity of each spectrum or the intensity at a selected wavenumber. Optionally, a white light image may be loaded for visualization, either manually or automatically.

### 2.1. Preprocessing

Atmospheric correction is the first selectable step in the preprocessing module. We developed a robust method for removing the contributions of gaseous H${}_{2}$O and CO${}_{2}$, described in detail in Methods. Figure 4a illustrates its application to a lignin spectrum which is heavily distorted by the presence of water vapor. In its basic form, the method merely subtracts a suitable amount of the atmospheric reference spectrum from the affected regions. Three options, active by default, are designed to eliminate the CO${}_{2}$ peak altogether and further remove residual water contributions. The method requires an atmospheric reference spectrum, preferably produced on the user’s spectrometer; a default spectrum is provided, covering 900–3840 cm${}^{-1}$ with 2 cm${}^{-1}$ resolution. As Figure 4a shows, the correction is comparable to, yet subtly different from, that of a common proprietary method.

Resonant Mie scattering correction can be performed as the next step, and will typically be the most time-consuming part of the preprocessing. The initial reference spectrum is important for guiding the correction process towards good spectra to explain the observed data. The user can provide a scattering-free spectrum, e.g., from transmission spectroscopy of a homogeneous sample, or choose among three provided spectra (casein, lignin and the Matrigel spectrum from RMieS-EMSC [11]).

To speed up the correction process, we developed an algorithm for clustered resonant Mie scattering correction, CRMieSC, which is explained in detail in Methods. In brief, the spectra are k-means clustered, an EMSC model is computed for each cluster, and the spectra are individually corrected using the least squares fit to the model. During this iterative process, the graph is continuously updated with the reference spectra of the clusters to visualize how convergence is reached. The number of clusters is selected in the graphical interface, and should be large enough that each cluster can consist of nearly identical corrected spectra. With our relatively homogeneous test images, 10 to 30 clusters proved enough to approximately reproduce the results of running the algorithm on each spectrum individually. With too few clusters, the corrected spectra will display spatial patterns corresponding to the clusters (not shown).

The ’Stabilize’ option initializes the process with a single cluster to quickly find a common reference spectrum, and convergence is improved by gradual updates of the reference spectra in the final iterations. The user can select the number of iterations, or use automatic stopping criteria as introduced by Solheim et al. [7]. Additional options, such as the choice of extinction curve model and associated parameters, are available through an ‘Advanced’ dialog. Clustering may be disabled to make the software correct the spectra individually, like in the RMieSC-EMSC and ME-EMSC algorithms. Figure 4b shows a randomly chosen spectrum from an image, corrected using different methods. The differences between ME-EMSC and CRMieSC with/without clustering are small compared with RMieS-EMSC; the main difference is a massive decrease in computation time.

Following atmospheric and scattering correction, a Savitzky–Golay denoising step [28] is available, in case the user desires to apply smoothing to the entire spectra. Spectra may further be limited to a specified wavenumber range, e.g., 900–1800 cm${}^{-1}$, where most biologically relevant information is typically found. Several background correction methods are available: rubberband, an iterated concave rubberband method, AsLS (asymmetric least squares) [29] and arPLS (asymmetric reweighted penalized least squares) [30]; these are primarily intended as an alternative to Mie scattering correction, e.g., for use on Raman spectra. Finally, spectra may be normalized by mean, L2-norm, maximum or intensity at a specific wavenumber, e.g., at the Amide I peak around 1650 cm${}^{-1}$. The preprocessing settings may be saved and loaded, both for convenience and record-keeping.

### 2.2. MCR-ALS Decomposition and Clustering

In the second part of the data processing pipeline, hyperspectral images are decomposed into concentration maps of a set of estimated ‘purest’ spectra, using the MCR-ALS method. The number of components to use in MCR-ALS is the main parameter for the user to select. As a guide, a singular vector decomposition (SVD) can be performed on the data; the eigenvalues are plotted and will give an indication of how much of the variability can be captured with a given number of components. After initialization of either the spectra or the concentrations using the SIMPle-to-use Interactive Self-modeling Mixture Analysis (SIMPLISMA) algorithm [31], the model is iteratively refined until the consecutive improvements in the residual errors have dropped below a user-specified threshold. As a fail-safe, the process also stops after a given number of iterations or if the error is found to be increasing.

Images sometimes contain regions that are irrelevant and disruptive to the analysis, caused by, e.g., bubbles or contamination in the material examined. These impurities affect the selection of initial spectra or concentration, and may mask relevant spectral differences in the studied material. In a small fraction of the test images, we observed that subsequent clustering was unable to segment the concentration maps into background and hyphae. Therefore, the program offers the option to manually select a region of interest (ROI) in order to exclude the disruptive regions of an image (Figure 3b). MCR-ALS is then applied only to spectra in the ROI; this procedure allowed the test images to be correctly segmented.

Following the MCR-ALS analysis, the spectra and concentration maps can be loaded in a separate clustering tool (Figure 3c), where the images are segmented using k-means clustering of the (optionally normalized) concentration values. Normalization allows the segmentation to reflect significant chemical differences across an image, in particular when large changes in some compounds would otherwise mask smaller changes in others. After the clustering step, the tool allows the user to manually annotate the resulting clusters, so as to inform further analysis steps about what spectra represent background, different types of cells etc. This annotation step allows the merging of several spectrally distinct regions into a single biologically relevant region. The average spectra of the clusters can then be visualized and exported.

### 2.3. Validation and Example Application

The software and new algorithms were validated on spectra from experiments where thin films of organic material (lignin) were colonized by the ectomycorrhizal fungus *Paxillus involutus*, as described in a separate publication [27]. Images of fungal hyphae were recorded during eight days of growth, under two slightly different growth conditions (with/without Fe${}^{3+}$ in the lignin film), each with six replicates. Each of the 96 image consisted of 64 × 64 spectra with 1530 absorbance values for wavenumbers from about 900 to 3850 cm${}^{-1}$. The speed of the software was evaluated on an HP EliteBook 840 G5 laptop computer with an Intel Core i7-8650U four-core processor, running openSUSE Linux.

Following atmospheric correction in OCTAVVS, the spectra were corrected for Mie scattering using CRMieSC with 30 clusters and up to 30 iterations, as well as with ME-EMSC in MATLAB with up to 50 iterations and no weighting of spectral regions. Correcting all 96 images took 50 min in CRMieSC and 33 h in ME-EMSC, a 40-fold speedup. For comparison, the original RMieS-EMSC method (with 30 iterations) can correct one image in about 7 h, requiring a month of computer time to process the whole experiment. These numbers are shown in Table 1.

The two sets of corrected spectra, as well as the raw spectra, were cut to 900–1800 cm${}^{-1}$ (467 absorbance values per spectrum) and then decomposed with MCR-ALS. The time needed to run MCR-ALS was found to vary considerably between images, depending on the number of iterations needed to reach the convergence criterion. With 8 components and the default settings of 700 iterations and termination at 0.1% improvement per iteration, the program took between 45 s and 8 min to decompose one image. In total, the process took 4$\frac{1}{2}$ h for each set of scattering corrected images but $6\frac{1}{4}$ h for the raw images.

Images were segmented using k-means clustering of the MCR-ALS concentrations, in order to identify background pixels and modified areas; see Figure 5a. There was good agreement between CRMieSC and ME-EMSC, and both methods considerably reduced the between-image variation compared with raw spectra (Figure 5b). Importantly, the chemical modifications around 1100 cm${}^{-1}$ were resolved only in the Mie scattering corrected spectra, demonstrating the value of this correction step in enhancing spectra, thereby aiding the identification of chemical species.

## 3. Discussion

By integrating commonly used methods for preprocessing and analysis of hyperspectral images from IR absorption microspectroscopy, OCTAVVS addresses a need to get tools for high-throughput data processing into the hands of biologists and other users of vibrational spectroscopy imaging. The graphical interfaces allow users to rapidly explore their options, choose suitable processing steps and apply those steps to a set of images.

Speed has thus been a major concern, in particular in the preprocessing steps where resonant Mie scattering correction was a major bottleneck until recently [7]. With the newly developed CRMieSC method, this scattering correction is no longer a time-consuming process. The total analysis time in our test case was in fact shortened by preprocessing with CRMieSC, as it allowed the MCR-ALS analysis to converge more quickly. Comparisons with ME-EMSC showed that CRMieSC produced similar results over a whole data set, although individual spectra may be negatively affected by the clustering procedure when combined with dynamic iteration control; future work will reduce such artifacts.

The aim to save time for the user also led to the development of a method for atmospheric correction in OCTAVVS, which was found to be comparable in quality to commercial software, but better documented and faster. There is still room for improvement, both in the handling of user-supplied atmospheric reference spectra and in how the water vapor contribution is modeled and removed, and the challenge is to create methods that require little to no manual adjustment.

The usefulness of the OCTAVVS MCR-ALS analysis tool for enhancing spectra and highlighting spectral changes was demonstrated on a set of 96 images, part of a larger study on the regulation of degradation processes in the ectomycorrhizal fungus *Paxillus involutus*. In that study, the MCR-ALS-ROI functionality was used to remove contaminated areas, and cluster annotation was performed to extract comparable spectra from all images [27].

OCTAVVS is aimed to provide an accessible graphical pipeline for a range of vibrational spectroscopy techniques. While we have applied the methods only to IR hyperspectral data, the MCR-ALS tool and several of the preprocessing steps would also be useful for Raman imaging. Future work will extend the software to cover additional algorithms and processing steps, e.g., noise reduction, binning, differentiation, PCA analysis and Raman spike removal. As other tools are readily available for further processing of the results with statistical or machine learning methods, the continued development of the OCTAVVS project will focus on tasks and methods that are specifically relevant to vibrational spectroscopy data.

Instructions for downloading and using OCTAVVS can be found at pypi.org/project/octavvs.

## 4. Methods

### 4.1. Atmospheric Correction

One simple method of atmospheric correction was described by Perez et al. [3]: The atmospheric spectrum is measured, e.g., as the difference between two spectra recorded from the same sample under different atmospheric conditions. The relative intensities at two manually selected wavenumbers are used to determine how much of the atmospheric spectrum will be subtracted, with some care taken to keep the baseline level of the corrected spectra. For the specific problem studied, this method compared favorably to the atmospheric compensation tool in the commercially available OPUS software [3].

The method implemented in OPUS is not openly documented, but it apparently uses a pre-recorded atmospheric spectrum and possibly an iterative optimization process, as it takes several minutes to correct a single image. Examination of corrected spectra reveals that the final step of this correction consists of running the spectra through a smoothing filter, with more aggressive smoothing in the water regions than in regions without atmospheric contributions. It is unclear why this method performed poorly against the method of Perez et al. in their specific case, but it might be that the built-in atmospheric spectrum in OPUS did not closely match the spectra recorded in the experiments.

A more powerful and accurate method for atmospheric correction was described by Bruun at al., based on the observation that recorded atmospheric spectra vary not only in intensity but also in shape. From 120 recorded spectra, a model of 2–3 components per spectral region was constructed in several steps using a combination of PCA and high pass filtering. Spectra were corrected through this model, using least squares regression on the second derivatives [2]. This procedure enabled accurate removal of atmospheric contributions to the measured spectra, but it required many calibration measurements in addition to analysis decisions that may be difficult to automate. It was noted that 99% of the variance was explained by the first PCA component [2], which suggests that a correction method based on a single atmospheric spectrum is sufficient for most purposes. Furthermore, the variability observed in atmospheric spectra hints that such a spectrum should be recorded under the same conditions as the samples that are to be corrected.

In the method we have implemented for atmospheric correction, each spectrum x is corrected according to
(1)$$x^{\prime }=x-\sum_{i}a_{i}r_{i},$$
where $r_{i}$ describes the atmospheric spectrum in region *i*, and the factors $a_{i}$ are chosen to maximize the smoothness of the corrected spectra by minimizing the sum of squares of the first derivative:(2)$$E_{i}=\sum_{j}{\left(\left(x_{j}-a_{i}r_{j}\right)-\left(x_{j-1}-ar_{j-1}\right)\right)}^{2},$$
which yields
(3)$$a_{i}=\frac{\sum_{j}\Delta x_{j}\Delta r_{j}}{\sum_{j}{\left(\Delta r_{j}\right)}^{2}},$$
where $\Delta x_{j}=x_{j}-x_{j-1}$. Computing this correction for the water spectrum in the regions 1300–2100 cm${}^{-1}$ and 3410–3850 cm${}^{-1}$ takes only a fraction of a second for 4096 spectra. As expected, we find a strong correlation between the correction factors for the two water regions (not shown). Bruun et al. used that finding to predict the correction in the lower-wavenumber region from the higher, whereas OCTAVVS corrects the two regions independently.

Some water peaks may still be visible in the corrected spectra due to the variation in peak intensities. To alleviate this problem without requiring multiple atmospheric spectra, we added an option to subsequently apply the correction as described above but computed in a narrower window. A window size of 60 cm${}^{-1}$ is used in the 1300–2100 cm${}^{-1}$ region, whereas 300 cm${}^{-1}$ was found to work best at 3410–3850 cm${}^{-1}$. This procedure is repeated 5 times, each time computing a local $a_{j}$ around point *j* and correcting the spectra as $x_{j}^{\prime }=x_{j}-\frac{1}{4}a_{j}r_{j}$.

Smoothing is optionally applied in the corrected regions. To reduce visible high-frequency noise without unduly altering peak shapes and amplitudes, we used a third-order Savitzky–Golay filter with a window size of nine points.

The CO${}_{2}$ contribution proved difficult to correct in our spectra, possibly because of greater variation in relative peak levels. As this region typically contains no biologically relevant peaks, the software offers a choice between correcting the region 2280–2430 cm${}^{-1}$ by the same method as the water regions (with a window with of 80 cm${}^{-1}$ for the additional correction step), or simply replacing the entire region with a non-overshooting spline.

Applying all these steps to an image with 4096 spectra takes about 10 s.

### 4.2. Clustered Resonant Mie Scattering Correction

The fundamental concepts of Mie scattering correction methods were described by Kohler et al., who used extended multiplicative signal correction (EMSC) [32] to formulate a model of the observed spectrum as the sum of a reference spectrum and a large set of extinction spectra generated by particles of different size and optical properties (specifically: spheres of different diameter and refractive index). A key step in this method is the use of PCA to reduce the extinction spectra to a small set of PCA loading vectors, with benefits to both stability and speed [4].

The index of refraction is connected to the absorption spectrum through the Kramers–Kronig relations, and this must be taken into account when possible extinction spectra are generated [5]. In the resonant Mie scattering correction algorithm (RMieS-EMSC) by Bassan et al. [11], the index of refraction is computed from the reference spectrum and used in the generation of extinction spectra. Like in the earlier method, a linear model is constructed for the observed spectrum in terms of the reference spectrum and scattering contributions. As the corrected spectrum better approximates the true absorption spectrum, the process is iterated with the corrected spectrum as the new reference spectrum [11]. The reference spectrum should, ideally, be free from scattering effects and be similar to the true absorption spectra. It is possible that the initial reference spectrum only weakly influences the corrected spectra [12], but it has also been observed that the reference spectrum can introduce strong biases [7].

After orthogonalization and PCA reduction (without normalization), the resulting EMSC model can be formulated as
(4)$$Z_{\mathrm{raw}}\left(\tilde{\nu }\right)=c+m\tilde{\nu }+hZ_{\mathrm{ref}}\left(\tilde{\nu }\right)+\sum_{i=1}^{N}g_{i}p_{i}\left(\tilde{\nu }\right)+\epsilon \left(\tilde{\nu }\right),$$
where $Z_{\mathrm{raw}}$ is the observed spectrum, $Z_{\mathrm{ref}}$ the reference spectrum and $p_{i}$ the *i*th PCA loading vector from the extinction matrix. The linear component $m\tilde{\nu }$ is optional, as noted below. After fitting the model parameters *c*, *m*, *h* and $g_{i}$ by least squares regression, the corrected spectrum is given by $Z_{\mathrm{corr}}=Z_{\mathrm{ref}}+\epsilon$ or, alternatively, $Z_{\mathrm{corr}}=\left(Z_{\mathrm{ref}}+\epsilon \right)/h$.

The exploration of a three-dimensional parameter space for the generation of extinction spectra makes RMieS-EMSC computationally costly. Mie scattering correction with RMieS-EMSC is impractically slow for large data sets, since a model of the extinction spectra is computed and iteratively refined for every input spectrum. Two major improvements by Konevskikh et al. reduce the computational complexity considerably: Computing the Kramers–Kronig relations through the Hilbert transform, and eliminating one of the three parameters used with the van de Hulst approximation of the extinction spectra [6,13]. These improvements have become more accessible with the recent publication of an open-source implementation in MATLAB of the ME-EMSC algorithm by Solheim et al. [7]. ME-EMSC drastically reduces the time needed to construct the model, but this process must still be carried out multiple times for every input spectrum. However, the first iteration is performed only once, as the whole image uses the same initial reference spectrum. This suggests a way to speed up the correction process: a group of highly similar spectra can be corrected using a common model.

We have developed a new algorithm for clustered resonant Mie scattering correction, CRMieSC. This algorithm has been implemented in Python, based in part on EMSC scripts developed by R. Perea Causín [33]. As briefly explained in Results, the spectra of an image are clustered with a k-means clustering, and the PCA loading vectors $p_{i}$ are computed for each cluster *k* from the mean of its constituent spectra, ${\langle Z_{\mathrm{ref}}\rangle }_{k}$, whereupon the spectra are individually corrected using Equation (Equation 4). The $Z_{\mathrm{ref}}$ term is that of the individual spectrum. The clustering step is repeated at each iteration, since clusters may initially be determined more by the scattering contributions than by the underlying spectra.

To improve convergence and robustness, the ‘stabilize’ option adds initial iterations with only a single cluster, causing the reference spectrum to be adapted to the mean spectrum. Furthermore, in later iterations, the reclustering is disabled to allow reference spectra to be updated only gradually, i.e., as ${\langle Z_{\mathrm{ref}}\rangle }_{k}^{\left(n+1\right)}=\frac{1}{2}{\langle Z_{\mathrm{ref}}\rangle }_{k}^{\left(n\right)}+\frac{1}{2}{\langle Z_{\mathrm{corr}}\rangle }_{k}^{\left(n\right)}$.

In OCTAVVS, the EMSC model can be computed using two different methods, approximating the methods of Bassan et al. [11] or Konevskikh et al. [6]. The difference between the two methods in this implementation is whether Mie extinction spectra are generated from three parameters (*a*, *b* and *d*) or two (the rescaled parameters $\alpha_{0}$ and γ), respectively. The distributions of values for $\alpha_{0}$ and γ differ from those used by Solheim et al. [7]; further investigation is needed to determine if a better choice of parameter values can improve the correction or reduce the required number of extinction spectra.

We note that the method of Konevskikh et al. generates a model that depends nonlinearly on the reference spectrum. This nonlinearity interacts with the rescaling of the corrected spectrum by *h* and appears to both decrease the stability of the method and make the result depend on the initial scaling of the reference spectrum. Future versions of the algorithm will address these issues.

Various implementation details differ between our Python code and the two MATLAB scripts. Even so, a single iteration of RMieS-EMSC or CRMieSC (without clustering) can produce highly similar results (not shown), provided that one major correction is made: the code for RMieS-EMSC appears to contain an erroneous multiplication by the wavenumber in the computation of the Kramers–Kronig transform, whereas ME-EMSC and CRMieSC are in agreement on this point.

Differences in speed between CRMieSC, RMieS-EMSC and ME-EMSC depend on settings such as the number of clusters used. With 30 clusters, CRMieSC took about 35 s to process an image of 4096 spectra; this required about 20 min with ME-EMSC and $7\frac{1}{2}$ h with RMieS-EMSC. Visual inspection of spectra corrected by CRMieSC revealed less variability in peak positions than among spectra corrected by RMieS-EMSC; it appears that noise is reduced by averaging many similar spectra before constructing the EMSC models.

### 4.3. MCR-ALS

The main idea of MCR-ALS is to decompose the measured spectral data matrix D into the purest concentrations matrix C and the purest spectra matrix S:(5)$$D={\mathrm{CS}}^{T}+E,$$
where E is the error or residual matrix. Equation (Equation 5) can be solved by iterative or non-iterative methods, but iterative methods are frequently used because of the flexibility to deal with diverse chemical problems [34].

The initialization of an iterative solver requires an initial estimate of either C or $S^{T}$. These can be obtained by different methods, such as SIMPLISMA [31] or independent component analysis (ICA) [35]. Of these, SIMPLISMA is the older and more frequently used method to extract spectral variation in IR and Raman spectra [19,36]. The main idea of this method is to extract the set of spectra or concentrations that best represent pure components in the chemical mixtures represented in the data. Our implementation of SIMPLISMA is based on MATLAB code by Jaumot et al. [26].

To solve Equation (Equation 5), OCTAVVS uses the pyMCR python library, which implements the alternating least squares (ALS) algorithm with non-negativity constraints on the spectra and concentrations [37]. In brief, Equation (Equation 5) is alternatingly solved for C and S using a non-negative least squares solver until the residual converges.

In MCR-ALS with region-of-interest selection (MCR-ALS-ROI), the user draws a polygon that defines what pixels of the image will be used. The spectra of the selected area are composed into a new data matrix, which is processed with MCR-ALS.

### 4.4. Cluster Mapping

As described in the main text, the output of MCR-ALS is imported into the graphical tool for clustering. Figure 6 shows an example of how the different components from MCR-ALS can correspond to different regions of an image, and how these components together determine the segmentation of the image. In this particular example from the fungal decomposition data set, three components capture the differences between the lignin film and the fungal hypha, while the fourth demonstrates the existence of a chemically distinct halo region around the cell. The relatively weak halo signal is amplified by clustering one the normalized concentration values, creating a separate cluster for the extended halo (Figure 6c).

## Figures and Tables

**Figure 1 mps-03-00034-f001:**
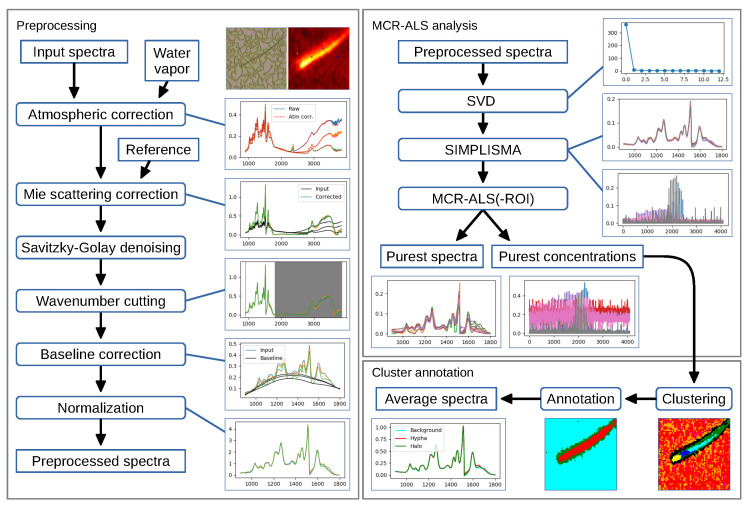
Preprocessing and analysis flowcharts. Left: Preprocessing steps in OCTAVVS, with examples of how an image can be corrected and transformed. Right: MCR-ALS analysis and clustering steps, using the preprocessed data as input and ending in the extraction of average spectra for three identified regions of the image. Example data from a fungal cell grown on lignin [27]. Rectangles represent data (sharp corners) or processing steps (rounded corners).

**Figure 2 mps-03-00034-f002:**
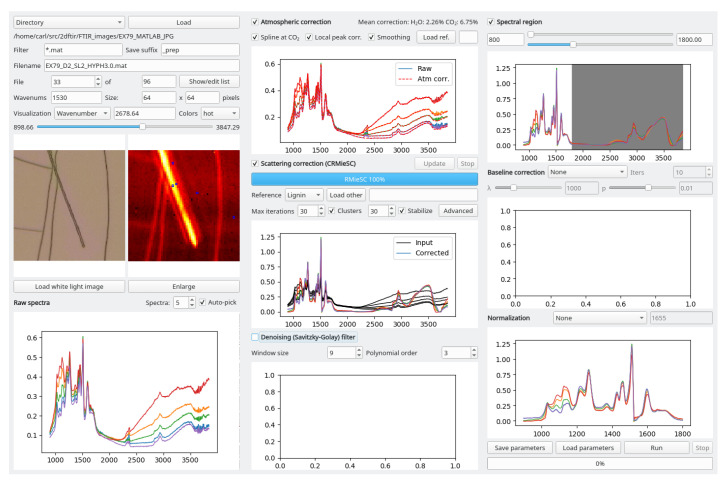
Graphical tool for preprocessing of vibrational spectroscopy images. The data flow follows the three columns, from left to right, beginning with the loading and visualization of input spectra. In this example, five representative spectra are visualized through a subset of the available processing steps. The figures, which may be enlarged, explored and saved, are automatically updated in response to parameter changes.

**Figure 3 mps-03-00034-f003:**
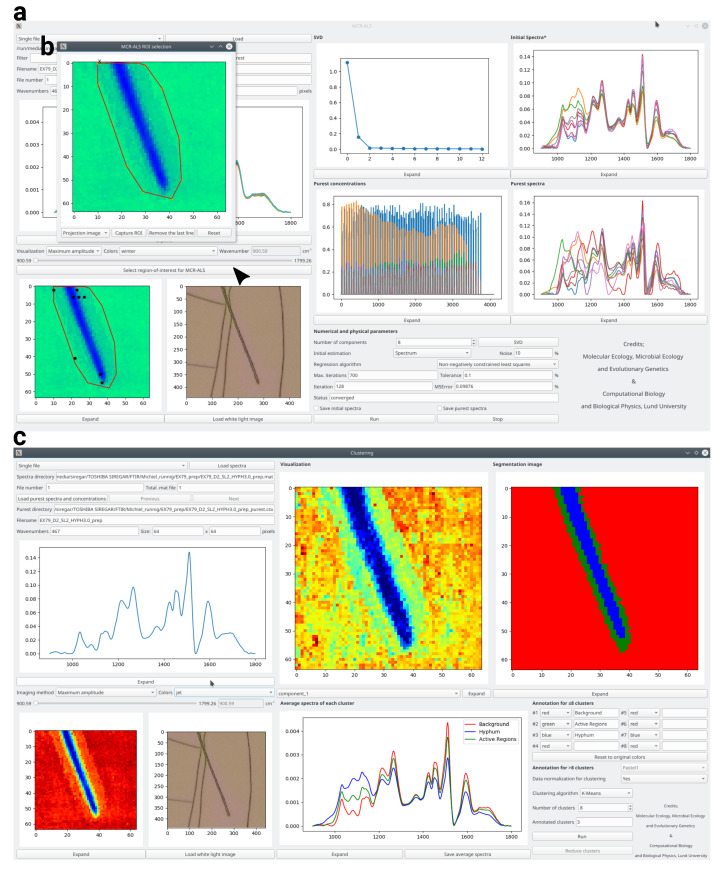
Graphical tools for MCR-ALS analysis and clustering. (**a**) MCR-ALS user interface, following decomposition of an image into pure concentrations (center) and spectra (center right). (**b**) Manual region-of-interest selection, here used to select a region around a fungal cell. (**c**) User interface for clustering, annotation and visualization. Eight clusters from k-means clustering (top center) are combined into three biologically relevant regions (top right) by manual annotation (bottom right), allowing mean spectra of the regions (bottom center) to be exported.

**Figure 4 mps-03-00034-f004:**
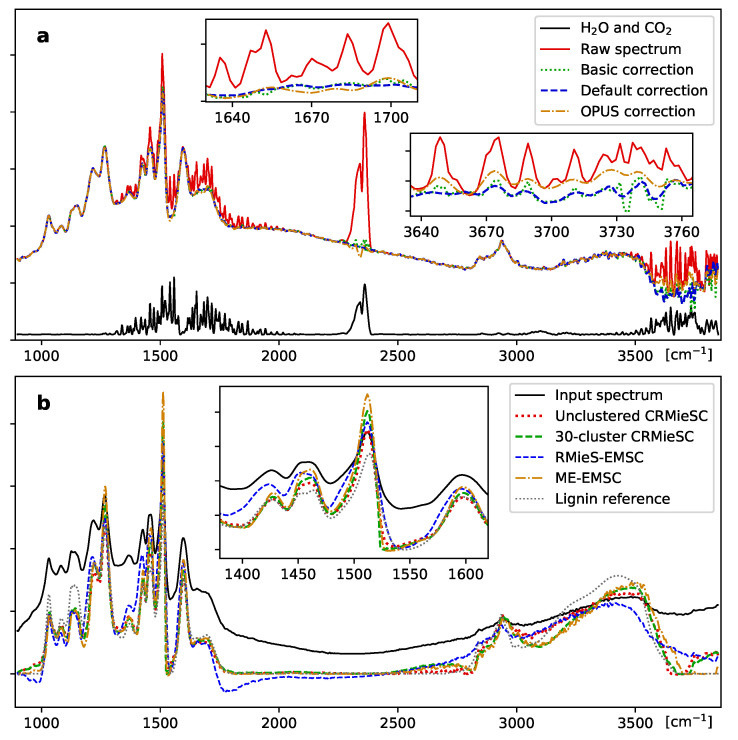
Atmospheric correction and Mie scattering correction. (**a**) The spectra of water vapor and CO${}_{2}$ (solid black line) can be clearly seen in a strongly affected lignin spectrum (solid red line). Correction by subtraction of the atmospheric spectrum in each affected region (dotted blue line) is compared with the full method described in the text (dashed green line) and the atmospheric compensation method in the OPUS software package (dash-dotted orange line). (**b**) Resonant Mie scattering correction, comparing the CRMieSC method with and without clustering of spectra (dashed green and dotted red lines) with the two previously published methods in MATLAB (dashed blue and dash-dotted orange lines). Insets show differences between the methods in selected regions. Settings: CRMieSC, 30 clusters, up to 30 iterations; ME-EMSC, no weights; RMieS-EMSC, 30 iterations, no Gaussian filter. Arbitrary units.

**Figure 5 mps-03-00034-f005:**
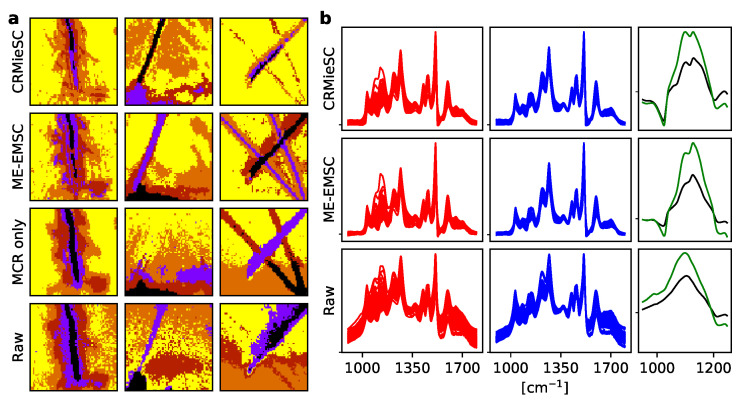
Example application: Lignin degradation by fungal hyphae. (**a**) k-means clustering (k = 5) of three typical images (columns) using normalized concentrations from MCR-ALS following atmospheric and Mie scattering correction with CRMieSC or ME-EMSC (first two rows) or MCR-ALS on the raw spectra (third row), compared with clustering the raw spectra (fourth row). Wavenumber range 900–1800 cm${}^{-1}$. (**b**) Mean foreground (red, left) and background (blue, middle) spectra of 96 images processed as in (**a**). The mean foreground–background differences are highlighted in the carbohydrate region (1000–1200 cm${}^{-1}$), separated by treatment (48 images each, black and green, right). Spectra were mean normalized before computing per-image means. Arbitrary units.

**Figure 6 mps-03-00034-f006:**
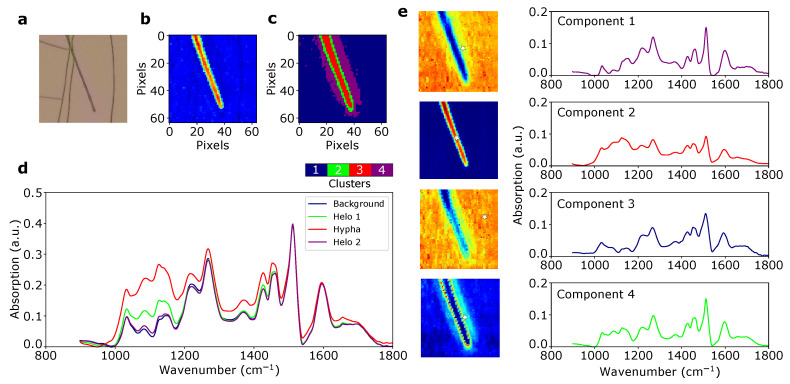
Clustering of MCR-ALS data. (**a**) White light image of a *Paxillus involutus* hypha on lignin film. (**b**) IR spectra of the same hypha, shown as total absorbance in the range 900–1800 cm${}^{-1}$. (**c**) Segmentation map of pure concentrations from MCR-ALS (with 4 components), obtained by k-means clustering. (**d**) Average spectra of each region in the segmentation map. (**e**) Concentration maps for the four pure components from MCR-ALS, and the associated spectral profiles (component 1–4). White star markers show the locations of the corresponding purest pixels.

**Table 1 mps-03-00034-t001:** Speed comparison between resonant Mie scattering correction algorithms. The algorithms were applied to a set of 96 images, as described in the text.

Method	Total Time	Relative Time
CRMieSC	50 m	1
ME-EMSC	33 h	40
RMieS-EMSC	28 day	800

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
