# Peer review of "OCTAVVS: A Graphical Toolbox for High-Throughput Preprocessing and Analysis of Vibrational Spectroscopy Imaging Data"

_mps, 2020, doi:10.3390/mps3020034_

Round 1

Reviewer 1 Report

This is a very well written manuscript presenting excellent science. I looked into the OCTAVVS tool box and found it very useful. However, I do have few questions. 

1) Does the software take in considering scattering by spherical and non spherical bodies? Does considers the refractive index of the liquid?

Only change I would recommend is to give mire details about the software used in toolbox. 

I look forward to seen this paper after minor changes.

Author Response

We thank the reviewer for these kind words.

1. The scattering correction is based on equations for homologous spheres. A few words have been added in Methods to clarify that the particles are spheres. In the scattering model by Bassan et al., one of the three parameters describes the refractive index of the material, and the other two the size and refractive index of the spheres. In the newer model (in the papers/code by Konevskikh and Solheim), one of the three parameters is eliminated but the equations are also different and we've found it difficult to really compare them, especially since we generally don't have the true unscattered spectra to compare with.

It is unclear to us to what extent other structures (such as the hyphae in our images) can be approximated as a mixture of spheres. But even with just spheres of different sizes and refractive indexes, there is the problem that just about any shape of the spectrum could be explained as scattering. That is, the mixture of particle properties gives enormous freedom; the compression of the extinction spectra by PCA is more important as a way of limiting the degrees of freedom than as an optimization for speed.

2. While more could have been said about details of the tools, we did not want the article to read too much like a manual. We tried to find a good balance between describing the software and describing the new algorithmic developments. The software itself is somewhat of a moving target, as we are committed to further develop it.

Reviewer 2 Report

The manuscript “OCTAVVS: A graphical toolbox for high-throughput pre-processing and analysis of vibrational spectroscopy imaging data” by C. Troein et al. introduces a new graphical toolbox for pre-processing and analysis of hyperspectral data (images). The toolbox is free, platform-independent and was developed with a variety of solutions incorporated to reduce the time (and computational power) required for pre-processing and analysis of large data sets. The authors present clearly the content of their toolbox as well as explain well the background behind each function. The toolbox contains basic, most commonly used options for data pre-processing (including e.g. atmospheric correction, Mie scaterring correction, denoising, normalisation, etc.) as well as MCR-ALS analysis. The manuscript contains also a well-presented and convincing example of the use of toolbox on data.

The manuscript is well-written, proving a clear and logical background for used functions. I fully agree with the authors that there is a need for user-friendly and platform universal graphical tool for processing and analysis hyperspectral images. In my opinion the authors deliver such a tool and it will be of great use for a broader scientific community (in particular, for mid-IR spectroscopist). Therefore, I do recommend the manuscript for publication in Methods and Protocols.

Author Response

We thank the reviewer for these kind words.

Some minor spelling errors etc have been corrected.

Reviewer 3 Report

Check the spelling of the article (some suggestions are added within the attached document).

Update some of the references with recent and relevant research.

All figures must contain axis labels to understand their analysis and the name of the graphics.

Review the structure of the article.

  • It looks like a review of the results, and if so, consider more papers that can support the research.
  • Tables are required to help understand comparisons of different results visually.
  • A chapter of conclusions is required where you can find a summary of the results and their evidence.

The contributions are not clear in the abstract, since many are added in the text and then the objective of the research is lost.

The time measurements were made in the simulations, but it is not explained how they will obtain these even numbers that mean years.

Can you explain your advantages of the software language vs others like LabVIEW or Qt. Certain references can be found in references:

A PC-based architecture for parameter analysis of vector-controlled induction motor drive

Author Response

> Check the spelling of the article (some suggestions are added within the attached document).

We thank the reviewer for these suggestions. We have adopted many of them and made changes where requested. We find no mention of the serial comma in the MDPI guidelines. As common style guides disagree on this point, we have regarded it as a matter of personal preference only.

We have followed the reviewer's suggestion to hyphenate 'open-source', but we have not changed 'least squares' as the cited papers spell it that way in their titles even when 'least squares' is used as an attribute.

> Update some of the references with recent and relevant research.

We would, if we knew which ones the reviewer considers to be outdated.

> All figures must contain axis labels to understand their analysis and the name of the graphics.

Assuming that this refers to figures 3 and 4, we have added [cm^{-1}] in figure 4 and a note in both legends to clarify that spectra are shown in arbitrary units.

> Review the structure of the article.
> It looks like a review of the results, and if so, consider more papers that can support the research.
> Tables are required to help understand comparisons of different results visually.
> A chapter of conclusions is required where you can find a summary of the results and their evidence.

We are not sure if these generic comments refer to this manuscript, as they seem to concern experimental work rather than a software package and new algorithms.

The Results section briefly describes the software (2), the implemented preprocessing steps (2.1), MCR-ALS+clustering (2.2) and validation on a data set (2.3). There is already a section called Discussion that puts the work into context, but the nearest thing to "evidence" of the results is the software itself and the validation in section 2.3.

The spectra in the figures would not be suitable to print as tables. We have now made a table of the time it took to scattering correct the data set in section 2.3; perhaps it is clearer than the text.

> The contributions are not clear in the abstract, since many are added in the text and then the objective of the research is lost.

The main point in the abstract is that "We here present a free and platform-independent graphical toolbox that allows rapid preprocessing of large sets of spectroscopic images". We have changed a few words to make it clearer that the algorithm for resonant Mie scattering is new.

> The time measurements were made in the simulations, but it is not explained how they will obtain these even numbers that mean years.

There is no mention of years in the manuscript. We do not understand what the reviewer is referring to.

> Can you explain your advantages of the software language vs others like LabVIEW or Qt. Certain references can be found in references:
> A PC-based architecture for parameter analysis of vector-controlled induction motor drive

The reference is a bit odd, as motor control is far from spectroscopy or imaging. But in any case: LabVIEW is a proprietary visual programming language that seems to be popular in some fields. It would be hardly be a better choice than MATLAB, where many relevant spectroscopy algorithms have already been implemented. In the introduction, we outline reasons why we have chosen Python over MATLAB for this project.

Qt is not a language, it is a GUI toolkit for C++, with bindings for various other languages. As mentioned in Results, we use PyQt5, which is a Python interface to Qt version 5.

Round 2

Reviewer 3 Report

all my concerns have been addressed